# Neurodegenerative Lysosomal Storage Disorders: TPC2 Comes to the Rescue!

**DOI:** 10.3390/cells11182807

**Published:** 2022-09-08

**Authors:** Sandra Prat Castro, Veronika Kudrina, Dawid Jaślan, Julia Böck, Anna Scotto Rosato, Christian Grimm

**Affiliations:** Walther Straub Institute of Pharmacology and Toxicology, Faculty of Medicine, Ludwig-Maximilians-University, 80336 Munich, Germany

**Keywords:** TRPML, TRPML3, TRPA1, TRPM2, TRPV2, BK, emphysema, lung injury, COPD, asthma, cystic fibrosis

## Abstract

Lysosomal storage diseases (LSDs) resulting from inherited gene mutations constitute a family of disorders that disturb lysosomal degradative function leading to abnormal storage of macromolecular substrates. In most LSDs, central nervous system (CNS) involvement is common and leads to the progressive appearance of neurodegeneration and early death. A growing amount of evidence suggests that ion channels in the endolysosomal system play a crucial role in the pathology of neurodegenerative LSDs. One of the main basic mechanisms through which the endolysosomal ion channels regulate the function of the endolysosomal system is Ca^2+^ release, which is thought to be essential for intracellular compartment fusion, fission, trafficking and lysosomal exocytosis. The intracellular TRPML (transient receptor potential mucolipin) and TPC (two-pore channel) ion channel families constitute the main essential Ca^2+^-permeable channels expressed on endolysosomal membranes, and they are considered potential drug targets for the prevention and treatment of LSDs. Although TRPML1 activation has shown rescue effects on LSD phenotypes, its activity is pH dependent, and it is blocked by sphingomyelin accumulation, which is characteristic of some LSDs. In contrast, TPC2 activation is pH-independent and not blocked by sphingomyelin, potentially representing an advantage over TRPML1. Here, we discuss the rescue of cellular phenotypes associated with LSDs such as cholesterol and lactosylceramide (LacCer) accumulation or ultrastructural changes seen by electron microscopy, mediated by the small molecule agonist of TPC2, TPC2-A1-P, which promotes lysosomal exocytosis and autophagy. In summary, new data suggest that TPC2 is a promising target for the treatment of different types of LSDs such as MLIV, NPC1, and Batten disease, both in vitro and in vivo.

## 1. Introduction

Recycling is the method of turning used and waste materials into new products. Recycling is highly relevant, not only in an economic or environmental context, but it is also an important principle in biology. From plants to animals, nutrient recycling by autophagy is a highly elaborated process to increase chances of survival during starvation periods, decrease dependence on external resources and efficiently reuse precious materials. Successful recycling requires proper “waste” management. Errors in or failure of this “waste” management not only result in a decreased ability to adequately recycle material but can lead to severe disease or even lethal defects. Usually, the most vulnerable cells of the body become the first victims, i.e., cells with a low turnover rate or cells that do not divide, such as neurons.

In mammals, the endolysosomal system, including autophagosomes and lysosomes, is the core unit of nutrient and material recycling, and more than 60 different diseases are known in humans called lysosomal storage disorders (LSDs), i.e., diseases in which mutations in certain endolysosomal proteins, enzymes or membrane proteins such as nutrient transporters or ion channels, resulting in the endolysosomal accumulation of macromolecules, proteins or lipids and glycolipids. Examples of such diseases are neuronal ceroid lipofuscinoses (NCL, Batten disease), mucopolysaccharidoses such as Hurler or Hunter syndrome, sphingolipidoses such as Fabry, Gaucher or Niemann–Pick-type C1 (NPC1) disease or mucolipidoses such as mucolipidosis type IV (MLIV). Neurodegeneration, developmental delay, intellectual disability, motor dysfunction, corneal clouding and vision loss, seizures, hepatosplenomegaly and premature death are common features of many LSDs.

Here, we will discuss the endolysosomal cation channels TRPML1, mutations of which cause MLIV, and TPC2 in the context of neurodegenerative diseases and their potential as targets for neurodegenerative lysosomal storage disease therapy. A particular focus will be placed upon the recently published work on TPC2 small molecule activation to treat Batten disease, NPC1 disease and MLIV [1]. 

## 2. Candidates for LSD Therapy: TRPML1 versus TPC2

TPC2 knockout mice accumulate material inside endolysosomes such as EGF/EGFR, PDGF or LDL cholesterol [2,3]; the latter was also observed in TPC2 siRNA-treated [1] or TPC2 inhibitor-treated cells (Figure 1A,B). In addition to the cholesterol accumulation upon TPC2 inhibitor treatment, mislocalization of the fluorescently labeled lactosylceramide (LacCer) was also observed, accumulating in lysosomal compartments (Figure 1C). NPC1 patient cells, e.g., fibroblasts, likewise accumulate cholesterol, and so do MLIV patient cells [1,4,5]. Lactosylceramide trafficking is affected not only in NPC1 and MLIV cells but also in NPA, GM1 gangliosidosis and Fabry cells [1,5,6,7]. Another sphingolipid, sphingomyelin, accumulates in NPC1 cells, and recently it was shown that Batten disease patients might also exhibit very high sphingomyelin levels [8]. 

Shen et al. (2012) [5] demonstrated that sphingomyelin blocks TRPML1 channels, suggesting that LSD patients with high sphingomyelin levels may benefit from the “reactivation” of TRPML1 channels with potent small molecule agonists. Thus, small molecule activation of TRPML1 has been shown to revert or rescue cellular phenotypes in NPC1 cells [5]. In addition, during activation of the lysosomal Ca^2+^-activated potassium channel (BK), TRPML1-dependently rescues aberrant lysosomal storage in NPA and Fabry disease [9], and loss of FIG4 (polyphosphoinositide phosphatase) and PYKfyve (FYVE finger-containing phosphoinositide kinase), which are both involved in the synthesis of the endogenous TRPML/TPC agonist PI(3,5)P_2_, is associated with neurological or neurodegenerative disease phenotypes [9,10,11] that can be rescued by TRPML1 activation [11]. Likewise, activation of mutated yet normally localized TRPML1 channels, causing MLIV with TRPML1 channel agonists, has been demonstrated to revert phenotypes in patient’s cells, while in cells from patients with complete loss of TRPML1 function (“knockouts” (MLIV2527 and MLIV2048); Figure 2A), treatment was ineffective. Patients with strongly mislocalized TRPML1 protein expression may likely benefit only to a very limited extent from TRPML1 agonist treatment (see, e.g., TRPML1(R403C) and TRPML1(V446L) [7] or TRPML1(T121M), Figure 2A,B). The latter TRPML1 mutant, T121M, was identified recently in an 18-year-old female MLIV patient of Yazidi origin in Germany (Figure 2). Using endolysosomal patch-clamp electrophysiology [12], we found the TRPML1^T121M/T121M^ channel to retain some residual activity in patient-derived fibroblasts when activated with the TRPML channel agonist ML-SA1 [5] or the TRPML1-selective agonist ML1-SA1 [13] (Figure 2C–F). Surprisingly, despite the severe reduction in channel activity and the apparent mislocalization, the patient presented with a fairly mild yet steadily deteriorating condition. 

While phenotypes such as cholesterol accumulation or LacCer mislocalization cannot be rescued with TRPML1 agonists in “knockout” MLIV patient cells or cells with dysfunctional or severely mislocalized mutant channels, mutants with largely correct localization (i.e., lysosomal localization) such as the TRPML1 variant F408Δ were found to respond to TRPML1 agonist treatment [7]. By contrast, all variants, including MLIV “knockouts” and mislocalized variants, were rescued with the TPC2 agonist TPC2-A1-P (Figure 2A). 

TRPML1 activity is highly pH-dependent, i.e., channel activity decreases with increasing lysosomal pH. Indeed, an increase in lysosomal pH is not unusual in LSDs, potentially hampering the accumulation of inhibitory lipids, as demonstrated for sphingomyelin and TRPML1 activity [14,15,16,17,18,19,20]. In contrast to TRPML1, TPC2 activity seems to be largely pH-independent (in the pH range of 4.6–7.4 demonstrated for humans and 4.6–6.0 shown for rabbit TPC2; [21,22]) and not blocked by sphingomyelin (Figure 3), possibly explaining the consistent efficacy of the TPC2 agonist (Figure 2A). In line with this, we also found reduced TRPML1 activity in Batten disease iPSC-derived cortical neurons (CLN3 knockout and CLN3^D416G^), the latter showing a severe clinical phenotype, while CLN3^R405W^ resulted in a retinitis pigmentosa phenotype, presenting with a fully active TRPML1 channel (Figure 4).

Functionally, both TRPML1 and TPC2 were shown to promote autophagy as well as lysosomal exocytosis, likely in a Ca^2+^ dependent manner as demonstrated and reviewed extensively in previous publications [1,23,24,25,26], with the potential advantage of human TPC2 activity being pH-independent and not affected by sphingomyelin accumulation. Thus, we concluded in a recent review that [26]: “Given the strong effect of TPC2 activation on lysosomal exocytosis when activated with the lipophilic PI(3,5)P_2_-mimetic TPC2-A1-P, it would be interesting to investigate how PI(3,5)P_2_-like TPC2 activation affects lysosomal function, autophagy, and cell viability in LSDs and ND”. Scotto Rosato et al. (2022) [1] addressed this question in selected LSD models, i.e., patient fibroblasts and iPSC-derived neuronal models for Batten disease, NPC1 disease and MLIV and in an in vivo model for MLIV.

## 3. TPC2-A1-P Rescues Phenotypes of Several LSDs In Vitro and In Vivo

In detail, we demonstrated that treatment with TPC2-A1-P in MLIV and NPC1 fibroblasts significantly recovers the lysosomal accumulation of LacCer. LacCer trafficking is influenced by intracellular cholesterol levels [5,6,7,27], and, in the case of cholesterol overload, it accumulates in endolysosomal compartments [28]. Therefore, in accordance with the lysosomal accumulation of LacCer, we also observed that both NPC1 and MLIV fibroblasts showed heavy cholesterol accumulation, which was efficiently reduced by TPC2 activation. While neither changes in LacCer trafficking nor cholesterol accumulation were detectable in Batten disease, the latter is known to accumulate lipofuscin [29] and globotriaosylceramide (Gb3) [30], and treatment with TPC2-A1-P was able to rescue both [1].

A common feature of LSDs is progressive neurodegeneration. To further corroborate the data from patient fibroblasts in a more relevant cellular model, iPSCs models, created using CRISPR/Cas9, were generated, carrying the most common mutation causing MLIV (MCOLN1IVS3-2A>G) or different Batten disease mutations (CLN3D416G (severe phenotype), CLN3R405W (mild phenotype) and CLN3ΔEx4-7). From these, cortical neurons were derived following established protocols [31,32]. We examined these neurons in direct comparison to isogenic WT neurons by analyzing lysosomal cathepsin B (CtsB) activity, LysoTracker (LyTr) staining and ultrastructures using electron microscopy. MCOLN1IVS3-2A>G neurons exhibited significantly increased CtsB activity as well as protein levels [33], and TPC2-A1-P treatment was able to significantly decrease both isogenic control levels. Furthermore, TPC2-A1-P was able to rescue lysosomal compartment expansion, highlighted by LyTr staining, in both MCOLN1IVS3-2A>G and Batten disease. In addition, using electron microscopy, we found lysosomal inclusion bodies in MCOLN1IVS3-2A>G neuronal progenitor cells (NPC), and their number was significantly decreased upon TPC2-A1-P treatment. No significant changes in lysosomal inclusion bodies were detected in Batten disease cells; however, the Cristae numbers per mitochondrial area were significantly reduced in CLN3ΔEx4-7, and TPC2-A1-P treatment significantly increased these numbers again.

Alongside the promising data observed in vitro, we took advantage of the well-established mouse model for MLIV [34,35] to test the pharmacological activation of TPC2 in vivo. The neuropathology observed in MLIV mice is characterized by early behavioral deficits, activation of microglia and astrocytes and, due to an impairment in protein degradation, P62/SQSTM1 aggregates accumulate in the central nervous system [34,36]. MLIV mice injected with TPC2-A1-P were found to show significant amelioration of the astrogliosis phenotype in the cerebellar arbor vitae, and a significant reduction of the number of P62/SQSTM1 aggregates in both the cerebellum and hippocampus. As an additional proof-of-concept, we tested TPC2-A1-P versus vehicle-treated mice on motor performance on the accelerating rotarod [37], demonstrating a significant rescue effect of TPC2-A1-P over vehicle treatment in MLIV mice.

Mechanistically increased lysosomal exocytosis and autophagy enhancement by TPC2-A1-P were shown. Both mechanisms were previously postulated to be potentially beneficial in treating different diseases [2,9,38,39,40,41,42,43,44], and recently we showed that TPC2-A1-P promotes lysosomal exocytosis in alveolar macrophages [45]. Accordingly, we found that TPC2-A1-P is also able to induce lysosomal exocytosis in LSD patient fibroblasts. In fibroblasts and iPSC-derived neurons, we also showed that treatment with TPC2-A1-P promoted starvation-mediated autophagy and P62/SQSTM1 clearance, suggestive of TPC2 being a modulator of autophagy under cellular stress conditions.

Altogether, these findings support the hypothesis that TPC2 activation by TPC2-A1-P is beneficial in neurodegenerative LSD treatment.

## 4. Is It All Clear Then?

No. A recent work by Tong et al. (2022) [46] claimed that exaggeration of TPC2 activity in SH-SY5Y neuroblastoma cells expressing mutant Presenilin 1 (PSEN1) and human fibroblasts from familial Alzheimer’s disease patients results in the reduction of lysosomal Ca^2+^, which in turn accelerates the Ca^2+^/H^+^ exchanger to expel H^+^ leading to lysosomal alkalinization and reduction in autophagy clearance of amyloids. Vice versa, inhibition of TPC2 by tetrandrine or Ned-19 or via siRNA treatment reportedly restored lysosomal Ca^2+^ and pH and rescued autophagy [46]. Albeit potentially different from the LSD models we investigated, the claim is that in AD patient fibroblasts and PSEN1 mutant neuroblastoma cells, blocking TPC2 would be beneficial compared to activation. However, no evidence, e.g., endolysosomal patch-clamp data supporting the “hyperactivity theory” of TPC2, was provided. Furthermore, these findings contrast results by Ambrosio et al. (2016) [47], who showed that a loss of TPC2 leads to an increase in pH in lysosome-related organelles (melanosomes), while activation of TPC2 by the PI(3,5)P_2_-mimetic TPC2-A1-P used in the study by Gerndt et al. (2020) was shown to have no effect on lysosomal pH [45]. On the other hand, activation of TPC2 by NAADP and the NAADP-mimetic TPC2-A1-N was indeed shown to result in the alkalinization of lysosomal pH [45,48,49]. Hence, the mode of activation may be critical, i.e., whether cells have aberrantly increased NAADP levels rather than increased PI(3,5)P_2_ levels. Such an aberrant TPC signaling hypothesis was also raised by Hockey et al. (2015) [50] when investigating molecular silencing of TPC2 or pharmacological inhibition of TPC signaling in fibroblasts from Parkinson’s disease patients with the common G2019S mutation in LRRK2. Therefore, the short-term blockage of NAADP-mediated TPC hyperactivation may correct the changes in pH driven by NAADP in such cases. Nevertheless, the long-term block or knockout of TPC2 seems rather problematic as this disrupts endolysosomal trafficking and degradation [2,3,51], suggesting that permanent inhibition might be counterproductive and would likely rather negatively impact neuronal health and survival. There is clearly a need to decipher the effects of different modes of TPC2 activation in neurodegenerative disease models, preferably in human iPSC-derived neuronal models, i.e., the differences in NAADP versus PI(3,5)P_2_-mediated channel activation, the possibility of NAADP mediated TPC2 hyperactivity in certain neurodegenerative disease models as well as long-term versus short-term inhibition of TPC2 in different disease models and disease conditions. Interestingly, two recent studies by Lee et al. (2010) [52] and Lie et al. (2022) [53] (same group) postulated a similar hyperactivity theory for TRPML1 in a PSEN1 model. Deacidification of endolysosomes after PSEN1 loss of function was claimed to induce pathological constitutive TRPML1 hyperactivation. Blocking TRPML1 channel activation reportedly reversed transport deficits in PSEN1 knockout neurons [53], while a role for TPC2, in contrast to Tong et al. [39], was excluded. As stated in a previous review [26], this claim of TRPML1 hyperactivity is not supported by direct TRPML1 channel activity measurements [54]. Endolysosomal patch-clamp evidence, by contrast, demonstrates that maximum activity of TRPML1 is achieved under acidic conditions, gradually decreasing with increasing pH [2,54,55,56]. How TRPML1 can be hyperactivated under elevated lysosomal pH conditions thus remains to be further scrutinized. One possibility might be a change in Ca^2+^ permeability of TRPML1 in a pH-dependent manner. The methodology to estimate the pH range used by all groups was LysoSensor Yellow/Blue-dextran. The main difference between the groups is the timing of pulse and chase and the different biological models used. Tong et al. [46] used a human neuroblastoma cell line and fAD fibroblasts while the Nixon group [52,53] and Coen et al. [57] used mouse cell lines, neurons isolated from PSEN1 KO mouse embryos and MEF, respectively. Due to the wide pH range present in endocytic organelles, addressing endolysosomal pH is a challenging task, especially when using endocytic tracers combined with pH sensitive dyes [58]. Therefore, it might be useful, to address these controversies using strategies and tools that have become available only recently such as the genetic lysosomal pH sensor pH-Lemon-GPI [45,59] or the DNA-based fluorescent reporter, CalipHluor, that can measure ratiometically luminal pH and Ca^2+^ [58].

## 5. Discussion and Conclusions

We conclude that activation of TPC2 with the PI(3,5)P_2_ mimetic TPC2-A1-P in fibroblasts from different LSD patients (Niemann–Pick-type C1, MLIV and CLN3-mediated juvenile Batten disease) as well as in neurons derived from iPSC models of MLIV and Batten disease results in the rescue of several disease phenotypes. These results are further backed up by in vivo data performed in an MLIV mouse model, such as the effects of TPC2-A1-P on astrogliosis, P62 accumulation in the cerebellum and hippocampus and motor performance (Rotarod experiments). Mechanistically, TPC2 activation may clear aberrantly accumulating material from lysosomes by lysosomal exocytosis. On the one hand, TPC2-mediated exocytosis is strongly promoted by TPC2-A1-P. In addition, TPC2-A1-P also promotes autophagy and may thus enhance the degradation of accumulated material in autophagolysosomes. In contrast to TRPML1, TPC2 only seems to promote autophagy under cellular stress conditions. On the other hand, potential benefits of TPC2 have been demonstrated, such as pH-independent activation by PI(3,5)P_2_ and its resistance to the sphingomyelin blockage, which both affect, by contrast, TRPML1 activity. Furthermore, dysfunction of TPC2 activity leads to increased levels of cholesterol, affecting intracellular trafficking [2]. These and other data [2] suggest that endolysosomal transport and degradation are impaired in TPC2-deficient cells. Data by Puri et al. (1999) [28] support these findings by demonstrating that cholesterol reduction restores proper trafficking of LacCer to the Golgi, whereas cholesterol overload redirects LacCer to endolysosomal compartments. Additional support comes from experiments where TPC2 was inhibited by different small molecule antagonists (Figure 1).

Despite the encouraging positive results with the TPC2 agonist TPC2-A1-P in some LSD models, it needs to be clarified if TPC2 activation might also be beneficial in other LSD and more common neurodegenerative disease models such as Parkinson’s or Alzheimer’s disease models. Hitherto, published data such as the ones in PSEN1 and LRRK2 models suggest that TPC2 inhibition may be beneficial [53]. The PSEN1 studies suggest an essential role for PSEN1 in the maturation and trafficking of the v-ATPase, responsible for lysosomal acidification. On the other hand, Coen et al. (2012) [57] and Zhang et al. (2012) [60] claimed that endolysosomal dysfunction in PSEN1 KO cells is not a consequence of failed N-glycosylation of V0a1 or compromised lysosomal acidification. Hence, both the role of PSEN1 in lysosomal acidification as well as the TRPML1 hyperactivity theory remain controversial. The studies by Tong et al. [46] and Hockey et al. [2] suggest hyperactivity of TPC2 and/or a beneficial effect of TPC2 inhibition that underscores the relevance of TPC2 in neurodegenerative disease conditions, but in these studies, it remains to be clarified what causes TPC2 hyperactivity, for example, excess NAADP. Direct channel activity measurements would be helpful in these models. In addition, Tong et al. [39], in contrast to the TRPML1 studies [45,46], showed data suggesting that TRPML1 is not involved in the phenotypes seen in PSEN1 cells. Vice versa, Lee et al. and Lie et al. [45,46] showed data to exclude the role of TPC2 instead. Finally, some of the tools used in the studies cited above are not sufficient to discriminate between TRPML1 and TPC2, such as Ned-19, which can block both channels. More selective tools are available and may be used to discriminate between these channels.

Many other open questions remain. What happens to aggregates and macromolecular material once exocytosed after TPC2-A1-P treatment? Will it be taken up and accumulated in other cells, or will it be degraded efficiently enough in the extracellular space? Are there other mechanisms beyond lysosomal exocytosis and autophagy stimulation that potentially contribute to the observed rescue effects? Would a combination of TRPML1 and TPC2 activation be beneficial? How critical is the mode of TPC2 activation (NAADP/TPC2-A1-N versus PI(3,5)P_2_/TPC2-A1-P)? Is TPC2 activation with TPC2-A1-P beneficial in adult-onset neurodegenerative diseases such as Alzheimer’s or Parkinson’s disease? What side effects would be expected? 

Despite these and other open questions, TPC2 appears to be a promising target for the therapy of certain LSDs, as discussed here, and it remains to be further investigated if and how additional LSDs and possibly even more common neurodegenerative diseases may benefit from TPC2 activation. 

## Figures and Tables

**Figure 1 cells-11-02807-f001:**
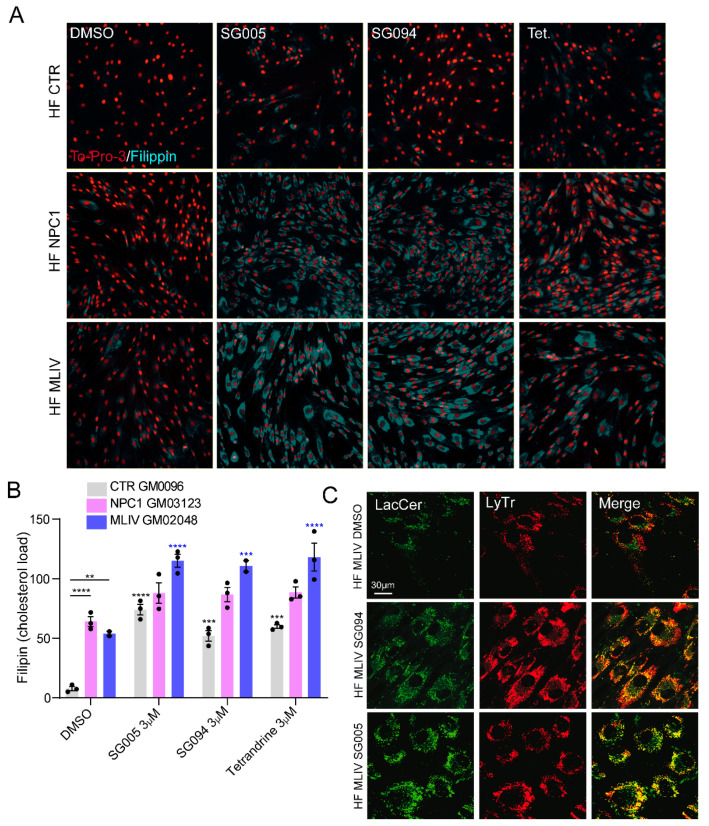
Effect of TPC2 inhibitors on cholesterol accumulation and lactosylceramide trafficking. (**A**) Representative confocal images showing filipin staining to visualize choles-terol accumulation and TO-PRO3 as nuclear staining. Experiments were performed in human fibroblasts: control (HF CTR), Niemann-Pick type C1 (HF NPC1) and mucolipidosis type IV (HF MLIV) treated with either DMSO, SG005 (3 µM), SG094 (3 µM) or tetrandrine (3 µM). (**B**) Bar plot showing the filipin intensity per cell (expressed as cholesterol load). Shown are mean values ± SEM. *n* > 3 biological replicates for each tested condition. * *p*-value < 0.05; ** *p*-value < 0.01; *** *p*-value < 0.001; **** *p*-value < 0.0001. Two-way ANOVA, post hoc Tukey’s multiple comparisons test. (**C**) Representative confocal images of LacCer (green) and LysoTracker (LyTr; red) in HF MLIV treated with DMSO, SG005 (1 µM) or SG094 (1 µM).

**Figure 2 cells-11-02807-f002:**
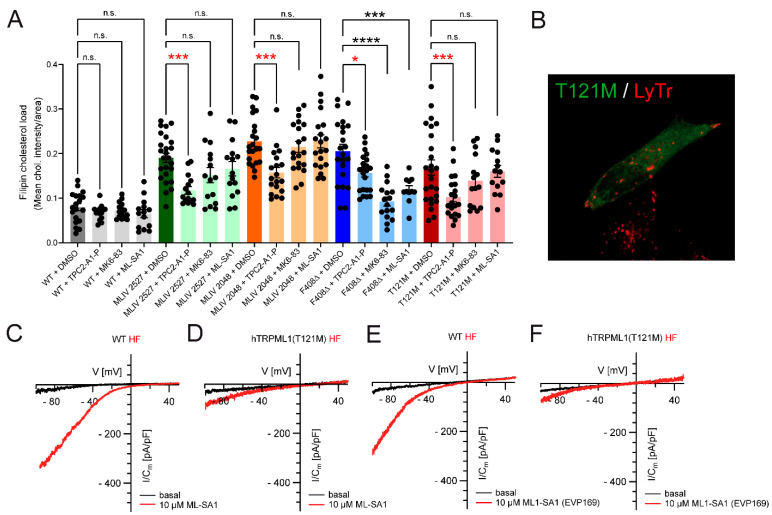
Effect of TPC2-A1-P on cholesterol accumulation in selected patient fibroblasts (HF) and electrophysiological characterization of a novel patient mutation (MCOLN1^T121M/T121M^). (**A**) Effect of TPC2-A1-P or the TRPML1 activators ML-SA1 and MK6-83 on cholesterol accumulation in WT, MLIV and selected patient fibroblasts, carrying MLIV-causing point mutations as indicated, visualized by filipin. MCOLN1^T121M/T121M^ is a severely mislocalized variant (**B**) found to be homozygously expressed in a patient from a Yazidi family, recently diagnosed with MLIV (Prof. Thorsten Marquardt, University of Münster, Münster, Germany). The patient, an 18-year-old woman, showed a comparably mild clinical phenotype for her age (ability to walk and talk, delayed development with retinal degeneration (risk of blindness), reduced iron (39 µg/dL (60–140)) and ferritin (6 µg/L (16–92)) levels, reduced Hb, HCT, MCV and MCHC (iron deficiency anemia), slightly deranged liver function (ASAT: 56 U/L (<30) and ALAT: 42 U/L (<30))). Electrophysiology revealed TRPML1^T121M/T121M^ to retain some residual channel activity. (**C**–**F**) Shown are representative currents (I-V traces) from vacuolin-enlarged LE/LY, isolated from WT or patient fibroblasts (HF), activated by ML-SA1 or ML1-SA1 (=EVP169 = selective TRPML1 agonist). * *p*-value < 0.05; *** *p*-value < 0.001; **** *p*-value < 0.0001.

**Figure 3 cells-11-02807-f003:**
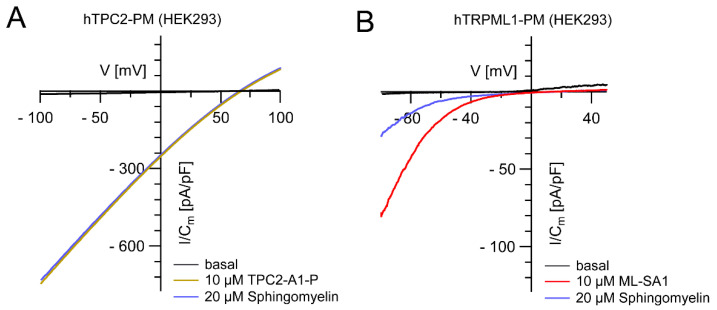
Effect of sphingomyelin on TPC2 and TRPML1 activities. (**A**,**B**) Representative measurements (I/Cm -V traces) of transiently transfected PM variants of hTPC2 (**A**) and hTRPML1 (**B**) in HEK cells. Channels were activated by small molecule agonists (10 µM TPC2-A1P and 10 µM ML-SA1), respectively. Subsequently, bath solution was completely exchanged for a solution containing agonist plus 20 µM sphingomyelin (SM). Bath solution contained: 138 mM NaCl, 6 mM KCl, 2 mM MgCl_2_, 2 mM CaCl_2_, 10 mM HEPES, and 5.5 mM D-glucose (adjusted to pH 7.4 with NaOH) and pipette solution contained 140 mM K-MSA, 5 mM KOH, 4 mM NaCl, 0.39 mM CaCl_2_, 1 mM EGTA and 20 mM HEPES (pH was adjusted with KOH to 7.2).

**Figure 4 cells-11-02807-f004:**
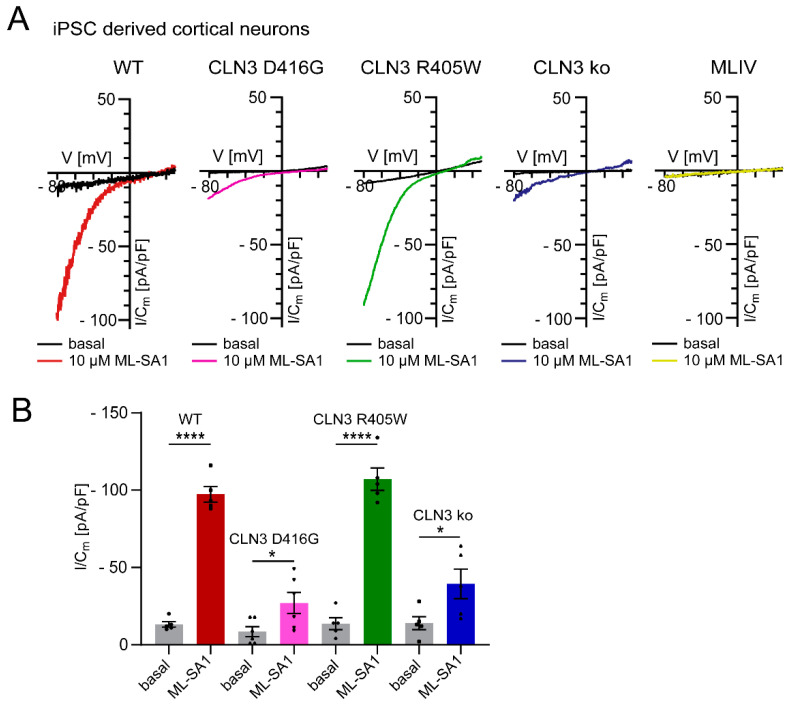
Effect of TRPML1 activation in different LSD and WT iPSC-derived cortical neurons (endolysosomal patch clamp experiments as described previously [12]). (**A**) Representative measurements (I/Cm -V traces) from apilimod-enlarged LE/LY, isolated from WT, CLN mutants or MLIV knockout iPSC derived cortical neurons activated with ML-SA1 (10 µM). (**B**) Statistical analysis of TRPML1 activation at −80 mV depicted as mean values ± SEM (WT, CLN D416G, CLN R405R or CLN ko; *n* > 5); each dot represents a single measurement from distinct neuronal differentiations. An unpaired *t*-test was applied to quantify statistical significance; * *p*-value < 0.05, **** *p*-value < 0.0001.

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
