# Peer review of "Neurodegenerative Lysosomal Storage Disorders: TPC2 Comes to the Rescue!"

_cells, 2022, doi:10.3390/cells11182807_

Round 1

Reviewer 1 Report

 In the manuscript entitled ‘Neurodegenerative lysosomal storage disorders: TPC2 comes to 2 the rescue!’ the author summarized and discussed recent development of TRPML1 and TPC2 with a focus on their roles in neurodegenerative diseases and potential therapeutic strategies to cure the diseases. The topic is interesting, timely, and significant. It was well written and easy to read.

Some suggestions for the authors to consider:

Line 24:

 ‘Although TRPML1 activation has shown rescue effects on LSD phenotypes’ should be ‘… has been shown to … phenotypes,’

Line 86:

Please double check ‘likewise activation of mutated, yet normally localized TRPML1 channels, causing MLIV with TRPML1 channel agonists was demonstrated to revert phenotypes in patients cells’

Line 213:

Please double check ‘Blocking TRPML1 activation reportedly re-213 versed transport deficits in PSEN1 knockout neurons [46] while a role for TPC2, in contrast to Tong et al. [39] was excluded.’

Author Response

We thank the reviewer for their supportive review and their very valuble comments. We have addressed their comments accordingly.

Reviewer 2 Report

The authors have addressed an interesting topic and covered some of the controversies in lysosomal cations and their role in neurodegeration. However, this discussion would benefit from more details of the techniques involved (especially when it comes to measuring lysosomal pH. At least the name of the probes used would be helpful in explaining some of the differences).

The authors state LacCer accumulates in a variety of diseases (line 72) but this is misleading, LacCer is not the major glycolipid to accumulate in these disease if at all and no method for the localisation of the natural lipid to lysosomes exists to the reviewer's knowledge. However, BODIPY LacCer trafficking is widely altered to lysosomes but the natural lipid does not widely accumulate. These two issues are widely confused throughout.

Altered Bodipy LacCer trafficking is due to the disruption in specific trafficking pathway in a cholesterol dependent this  is not referenced consitently and is presented in a confusing manner- pagano re should be referenced throughout - at least using reference 21 consistently

Line 108 no references are provided for the statement that lysosomal pH is widely disrupted in the LSDs examples such as NPC1 Wheeler et al NBD, 2019 and also Gaucher - Sillence, 2013; De La Mata et al., 2017; Magalhaes et al., 2015; Bourdenx et al., 2016; Chakraborty et al., 2017 should be cited. Tharkeshwar, et al., 2017. . Sci. Rep. 7, 41408. also showed increased pH in NPC1 in the figures.

Line 179 I suggest that you try 'Increased TPC2 activity'

In the controversy over changes in lysosomal pH (lines 180-194) it would be useful if the authors briefly describe the techniques involved and whether they are likely measuring similar compartments between studies as well as in healthy vs disease/treated? It's quite expected that different techniques will vary (eg a general acidic probe which would likely measure the whole of the endolysosomal compartment vs a specific mature lysosome probe). It should be noted that at least in NPC1 disease complete endosomal subpopulations are absent eg Chakraborty et al., 2017.

Author Response

We thank the reviewer for their supportive review and their very valuble comments. We have addressed their comments and made changes regarding LacCer statement, pH measurements (here we mention now the methods used and discuss alternative methods which recently have become available that may be used to reassess pH), and we have inserted the suggested literature.